# Non-Contact Measurements of Electrocardiogram and Cough-Associated Electromyogram from the Neck Using In-Pillow Common Cloth Electrodes: A Proof-of-Concept Study

**DOI:** 10.3390/s21030812

**Published:** 2021-01-26

**Authors:** Akira Takano, Hiroshi Ishigami, Akinori Ueno

**Affiliations:** 1Master’s Program in Electrical and Electronic Engineering, Graduate School of Engineering, Tokyo Denki University, Tokyo 120-8551, Japan; 20kmj23@ms.dendai.ac.jp (A.T.); 19kmj03@ms.dendai.ac.jp (H.I.); 2Department of Electrical and Electronic Engineering, Tokyo Denki University, Tokyo 120-8551, Japan

**Keywords:** cough-associated EMG, textile electrodes, capacitive ECG, non-contact capacitive measurement, bootstrapping technique, heart rate from the neck

## Abstract

Asthma and chronic obstructive pulmonary disease are associated with nocturnal cough and changes in heart rate. In this work, the authors propose a proof-of-concept non-contact system for performing capacitive electrocardiogram (cECG) and cough-associated capacitive electromyogram (cEMG) measurements using cloth electrodes under a pillowcase. Two electrodes were located along with the approximate vector of lead II ECG and were used for both cECG and cEMG measurements. A signature voltage follower was introduced after each electrode to detect biopotentials with amplitudes of approximately 100 µV. A bootstrapping technique and nonlinear electrical component were combined and implemented in the voltage follower to attain a high input impedance and rapid static discharge. The measurement system was evaluated in a laboratory experiment for seven adult males and one female (average age: 22.5 ± 1.3 yr). The accuracy of R-wave detection for 2-min resting periods was 100% in six subjects, with an overall average of 87.5% ± 30.0%. Clearly visible cEMGs were obtained for each cough motion for all subjects, synchronized with reference EMGs from submental muscle. Although there remains room for improvement in practical use, the proposed system is promising for unobtrusive detection of heart rate and cough over a prolonged period of time.

## 1. Introduction

Cough is the most common symptom for which patients seek medical advice [1,2,3]. Nocturnal cough can occur in asthma [4,5], chronic obstructive pulmonary disease (COPD) [6,7,8], obstructive sleep apnea [9,10,11], and gastroesophageal reflux disease (GERD) [12,13], among others. In addition, nocturnal morbidity and mortality are reported for asthma and COPD [14,15,16]. Nocturnal GERD is considered to be more harmful than daytime GERD [13,17]. As subjective assessments of nocturnal cough frequency are unreliable [18,19], cough detection devices are required for reliable objective assessments and diagnoses. Meanwhile, recordings must be obtained over a prolonged time period for cough detection due to the paroxysmal nature of coughs [20]. Therefore, a non-intrusive approach is desired for cough detection.

The diseases mentioned above are associated not only with cough but also with changes in heart rate (HR) or HR variability [21,22,23,24,25]. Pneumonia presents with both cough and increased HR [26,27]. Hence, a non-obtrusive method for measuring nocturnal cough and HR may be useful for diagnosing and assessing these diseases.

HR is generally calculated from the interval of R-waves (RRI) in electrocardiogram (ECG) measurements obtained with adhesive electrodes. In addition, some methods have been developed for cough detection. Previously reported methods include the use of microphones [20,28,29,30,31,32,33,34,35,36], video recordings [37], electromyography (EMG) [38,39], and accelerometers [40]. In routine cough measurements over a prolonged time period, however, audio sensing using a microphone or video recording may undermine the patient’s privacy [40,41]. Prolonged attachment of the adhesive electrodes used in existing ECG and EMG measurement devices can cause skin reactions or inflammation as the measurement period increases. Therefore, new methods for long-duration measurements of HR and cough are desirable to ensure broad acceptance among users.

One possible approach is the use of dry cloth electrodes via clothing for ECG and/or EMG measurements. Some researchers have successfully performed ECG [42,43] or EMG [44,45] measurements in such a way without direct skin contact. This approach can simultaneously reduce the risk of inflammation and privacy invasion. For cough detection, the laryngeal muscles [46] and surrounding muscles can be a target for EMG measurement. For HR detection, the R-wave in ECG has been shown to arise in the EMG signal obtained from hyoid muscles during polysomnography (PSG) [47]. Therefore, both EMG and ECG can potentially be measured from the neck region, neighboring the above described muscles.

Pillows are widely used during sleep almost daily, and the pillowcase is in contact with the posterior neck region over a long time period. Thus, dry cloth electrodes under the pillowcase may detect both cough-associated EMG and ECG in a contact-free manner over a long period of time without invading privacy. Based on these considerations, we developed and tested a prototype non-contact system for measuring cough-associated EMG and ECG from the neck using in-pillow cloth electrodes as a proof-of-concept study.

## 2. Materials and Methods

### 2.1. In-Pillow Cloth Electrodes

The cloth electrodes shown in Figure 1 were designed for passively measuring both the electrocardiographic and electromyographic signals from the neck. The measurements are mediated by capacitive couplings of the neck skin (conductor), pillowcase (insulator), and electrode (conductor). Accordingly, the alternating (i.e., time-varying) ionic displacement in the body (ECG and EMG signals) is converted to alternating electronic displacements without direct skin–electrode contact [42,43,44,45]. Each electrode plays a dual role of electrode and coupling capacitor at the front-end of the measurement circuit. Measurements by this approach are realized when the coupling capacitances are sufficiently large (meaning that the capacitive reactance of the coupling is sufficiently small), or when the input impedance of a front-end circuit connected to each electrode is sufficiently high to suppress the voltage loss of each coupling. Hereafter, the ECG and EMG signals measured via capacitive couplings are referred to as cECG and cEMG, respectively.

Sensing electrodes, depicted as (−elec.) and (+elec.) in Figure 1a, were arranged diagonally along with an approximate vector of lead II ECG according to the results of the following preliminary experiment, which was conducted on three subjects. In the preliminary experiment, four contact-type electrodes (F-150S, Nihon Kohden) were attached at the corner of the quadrangular posterior surface of the neck of the subject. Two sets of potential difference between the diagonal electrodes were simultaneously measured using a commercially available electrocardiograph (ECG100C, BIOPAC Systems). One set (Figure 1a) was arranged as described by Walter et al. [48] so that the detected electrical vector roughly followed the lead II ECG vector. The other set used the remaining two electrodes for measuring the flipped electrical vector. The amplitudes of ten R-waves of the obtained ECG were measured for each vector and compared among the vectors of each subject. The electrode layout of the prime experiment was that in Figure 1a, because in this layout, the amplitude of the R-waves was significantly larger than that obtained using the flipped layout (as confirmed by significance tests in all subjects). The common use of sensing electrodes for both cECG and cEMG reduced the required number of electrodes and increased the distance and size of the incorporated electrodes.

As shown in Figure 1b, a five-layered structure with a sensing layer, a shield layer and a ground (GND) layer was adopted [49,50]. Each sensing electrode was connected to a front-end voltage follower (FVF) as shown in Figure 2a. The output of the FVF (i.e., sensed voltage) was fed back to the driven shield layer via a 150-Ω resistor. The driven shield layer performs not only as a shield but also as a protector against current leakage from the sensing layer to the GND layer, decreasing the stray capacitance between the sensing and GND layers and increasing the common-mode rejection ratio (CMRR) [51]. Furthermore, the driven shield layer is known to protect the sensing layer from undesirable current caused by moving artifacts and electrostatic disturbances, thereby improving the artifact tolerance [49,50].

To prevent sleep disruption, the electrode is constructed from a thin, soft conductive fabric (CSTK, Kitagawa Industries) and insulating textiles (100% polyester), with a total thickness of 0.45 μm. The electrodes were fixed on a pillow (stuffed with 100% latex cushioning) to be placed at the neck and were covered with a cotton pillowcase (308 μm thick). When the patient lays their head and neck on the pillow, the electrodes form capacitive couplings with the neck via the pillowcase.

### 2.2. Measurement System

Figure 2a shows a block diagram of the constructed measurement system and Figure 2b illustrates the function of FVF in capacitive biopotential measurements. The source biopotential $e_{s}$ is divided by the impedance of the coupling capacitance ${\dot{Z}}_{cap}$, and the input impedance of FVF ${\dot{Z}}_{in\_FVF}$. The divided voltage becomes an input (output) of the FVF $v_{in\_FVF}$ ($v_{out\_FVF}$) as follows:(1)$$\frac{{\dot{Z}}_{in\_FVF}}{{\dot{Z}}_{cap}+{\dot{Z}}_{in\_FVF}}e_{s}=\frac{1}{{\dot{Z}}_{cap}/{\dot{Z}}_{in\_FVF}+1}e_{s}=v_{in\_FVF}\left(=v_{out\_FVF}\right).$$

When ${\dot{Z}}_{in\_FVF}$ is sufficiently larger than ${\dot{Z}}_{cap}$ (i.e., ${\dot{Z}}_{in\_FVF}\gg {\dot{Z}}_{cap}$), a voltage close to the source voltage is input to the FVF.

The FVF usually employs a simple voltage follower with high input impedance. However, there is a prolonged period during which the voltage input is out-of-range (for example, by static electricity with amplitudes exceeding 100 V). During this time, measurements are not possible. To achieve compatibility between the high input impedance of signal inputs with amplitudes of approximately 100 µV and the low input impedance of out-of-range voltage inputs, a bootstrapped varistor was incorporated in the FVF (see Figure 2c). The bootstrapping component supports a high input impedance when the input voltage is lower than the varistor threshold. When the input voltage exceeds the threshold, the input impedance of the varistor drops and the discharge time is reduced [52,53].

Subsequent circuits for measuring the cECG and cEMG signals consisted of an instrumentation amplifier, filters, and an inverting amplifier. The passband and gain were set to 0.5–100 Hz and 60.0 dB, respectively, for the cECG measurement and 40–1000 Hz and 72.0 dB, respectively, for the cEMG measurement. The cEMG component below 40 Hz was attenuated to suppress R-waves in the cECG.

### 2.3. Experimental Methods

All experimental procedures were approved by the Human Life Ethics Committee of Tokyo Denki University. All subjects provided informed consent prior to participation in our experiments. Eight healthy subjects (seven males, one female) participated in the experiment, as listed in Table 1. Subjects with long hair were asked to expose the skin on their neck. All subjects were instructed to lay their head on the pillow so that the electrodes were positioned beneath bare neck skin, as shown in Figure 2a. Prior to the experiment, subjects practiced the action of “coughing twice” with an interval such that the corresponding submental EMG (EMG_ref_) trains could be distinguished into two groups. Then, the subjects remained in a resting state in the supine position until the noise in the cEMG signal was attenuated.

The experimental protocol is shown in Figure 3. After the first rest period, the subjects were instructed to “cough twice” seven times as they trained. The time stamp for each “two-cough” cycle was recorded for reference. When a natural cough occurred after the last (i.e., seventh) two-cough cycle, the initiation of the second rest period was postponed to the end of the natural cough. cECG and cEMG signals were recorded with the proposed system throughout the entire experiment. As a reference ECG signal (ECG_ref_), the lead II ECG was simultaneously measured from the chest with a commercially available telemetry system (BN-RSPEC, BIOPAC Systems) and contact-type electrodes (F-150S, Nihon Kohden). The EMG_ref_ signal was simultaneously measured from the submental by reference to PSG [47] with a commercially available EMG amplifier (BA1104M, TEAC Instruments), telemetry system (TU-4, TEAC Instruments), and the same type of electrodes. Measured cECG and cEMG signals, as well as reference ECG_ref_ and EMG_ref_ signals, were digitized at 10 kHz with 16-bit resolution between −10 and +10 V by a commercially available A/D converter (BIOPAC Systems, MP150) and recorded on a personal computer.

### 2.4. Analytical Metheds

#### 2.4.1. Analysis of ECG Signals

The following analysis was applied to a 2-min segment before the first two-cough cycle. Moving averaging (20-ms time window), bandpass filtering (IIR, 10–20 Hz), and differential processing (10-ms interval) were applied to both ECG_ref_ and cECG segments as preprocessing. Five assumed R-waves were selected in a random manner from each preprocessed signal, and their mean amplitude was calculated. For R-wave detection, 80% of each mean amplitude was used as the threshold through the segment. The RRI was calculated from two consecutively detected R-waves. If the RRI of a cECG signal was within ±10 ms of the RRI of ECG_ref_, the latter R-wave of the cECG was deemed as correctly detected. The sensitivity (*P_SNS_*), accuracy (*P_ACC_*), and positive predictive value (*P_PPV_*) of the detected R-waves in cECG measurements were calculated by (2), (3), and (4), respectively:(2)$$P_{SNS}=\frac{N_{TP}}{N_{TP}+N_{FN}}\times 100,$$
(3)$$P_{ACC}=\frac{N_{TP}+N_{TN}}{N_{TP}+N_{TN}+N_{FN}+N_{FP}}\times 100,$$
and
(4)$$P_{PPV}=\frac{N_{TP}}{N_{TP}+N_{FP}}\times 100,$$
where *N_TP_*, *N_FP_*, and *N_FN_* are the numbers of R-waves correctly detected, falsely detected, and undetected, respectively. As there are no countable valid events corresponding to *N_TN_*, *N_TN_* was set at zero when calculating the accuracy [54].

#### 2.4.2. Analysis of EMG Signals

To verify that cough-associated cEMG measured from the posterior neck was synchronized with the EMG_ref_ obtained from submental muscle, the following analysis was performed. First, bandpass filtering (IIR, 40–500 Hz) was applied to both EMG_ref_ and cEMG signals as preprocessing. The analysis segment of each two-cough cycle in EMG_ref_ and cEMG was determined as shown in Figure 4. For each two-cough cycle in the EMG_ref_ signal, the segment (i.e., beginning and ending times) and its middle point were identified with reference to the recorded time stamp of the two-cough cycle, and the duration of the two-cough segment was measured (step 1). The first and last (seventh) two-cough segments were excluded for stability purposes, and the mean duration (MD) of the remaining segments was calculated for each subject (step 2). Extended segments with a duration 1.2-fold longer than the MD were selected for discrete-time Fourier transform (DFT) analysis (step 3). Hereafter, the extended segment with a common duration is called the *coughing segment*. Coughing segments in the cEMG signal corresponding to those in the EMG_ref_ signal were also selected for each subject. A DFT was applied to each coughing segment in the EMG_ref_ and cEMG data. Five stable resting segments with the same duration as the coughing segment were selected from the 2-min first rest period of the EMG_ref_ and cEMG signals, and were also analyzed by DFT. In the selection of the resting segment, any segments with visible EMG firings due to swallowing or inhalation were excluded.

The total spectral amplitude (TSA) from 40 to 500 Hz was computed in each DFT spectrum obtained from a segment. The TSAs were averaged over five segments of resting and five segments of coughing in the EMG_ref_ and cEMG signals of each subject. The mean TSAs between resting and coughing in the EMG_ref_ and cEMG signals were then compared for each subject, and the comparisons were quantified in a significance test. Based on the normality results, the EMG_ref_ and cEMG signals were evaluated by a paired *t*-test and a Wilcoxon signed-rank test, respectively.

## 3. Results

### 3.1. Simultaneous Measurements of cECG and Cough-Associated cEMG

Figure 5a shows example recordings of ECG_ref_, cECG, EMG_ref_, and cEMG, including the fifth and sixth two-cough segments obtained with the reference or proposed systems from subject B. Additionally, filtered and differentiated cECG signals are also depicted. Clearly visible periodic R-waves were observed in the cECG signal during the stable period, which were synchronized with those of the ECG_ref_ signal, as shown in Figure 5b. Although R-waves were unclear during and after the two-cough cycle due to fluctuations in the signal baseline, the timings of R-waves could be detected by applying a bandpass filter and differentiation to the cECG signal (see the third stage in Figure 5a). Similarly, most of the R-wave timings after the two-cough cycle were detected in all subjects except subject E. Synchronized R-waves similar to those in this figure were confirmed in all subjects except subject E. For cEMG signals, EMG firings during the two-cough period were synchronized with those in the simultaneously measured EMG_ref_ signal, as shown in Figure 5c. These synchronized firings were observed for all two-cough cycles in all subjects. Firings considered to be associated with inhalation and swallowing were also confirmed before and/or after the two-cough cycles in all subjects except subjects D and H.

### 3.2. Evaluation of cECG and cEMG Signals Measured in the Proposed System

#### 3.2.1. Evaluation of cECG Signals

The *P_SNS_*, *P_ACC_*, and *P_PPV_* values of R-waves in the cECG signals for each subject are given in Table 2, with averages of 90.7%, 87.5%, and 89.0%, respectively. Aside from those for subjects D and E, the *P_SNS_*, *P_ACC_*, and *P_PPV_* values were all 100%. In subject E, the R-wave amplitudes were considerably smaller than those of the other subjects, as shown in Figure 6. Consequently, approximately 85% of the waves were not detected by the current detection algorithm, resulting in a low *P_SNS_* (14.3%). Moreover, the number of falsely detected (*N_FP_* = 154) and undetected (*N_FN_* = 79) waves was substantially higher than that of correctly detected waves (*N_TP_* = 39), because many T-waves had a larger amplitude than the R-waves and were falsely detected as R-waves (see Figure 6).

#### 3.2.2. Evaluation of cEMG Signals

Figure 7 shows DFT spectra of the analyzed segments in the EMG_ref_ and cEMG signals depicted in Figure 5c. When comparing the target frequency components (40–500 Hz) of EMG_ref_ and cEMG, the spectral amplitudes for the coughing segment of both signals were higher than those of the resting segment. This increase in spectral amplitude for the coughing segment was observed in the cEMG signal for all subjects.

Figure 8a compares the TSAs of the resting and coughing segments, respectively, in the EMG_ref_ and cEMG signals of each subject. Consistent with the spectral amplitudes of each subject, the mean TSA values were higher during coughing than during resting in both the cEMG and EMG_ref_ signals of each subject. Figure 8b compares the group average of the mean TSAs over all subjects during resting and coughing in the EMG_ref_ and cEMG signals. As evidenced in the figure, the mean TSA was significantly higher in the coughing segments than in the resting segments, in both the EMG_ref_ signals (*p* < 0.01) and cEMG signals (*p* < 0.01).

As the mean TSAs were higher in the coughing segments of EMG_ref_ in all subjects (Figure 8a) and the difference was significant (Figure 8b), we confirmed that the segments segregated the periods including cough-associated muscle activities. Moreover, the mean TSAs were significantly increased in the coughing segments during the same periods of cEMG (Figure 8a,b), indicating that the proposed system detected cough-associated muscle activities via the pillowcase holding the posterior neck.

## 4. Discussions

### 4.1. Advantages of the Adopted Electrode Configuration and Novel FVF

As shown in Figure 5, the proposed system succeeded in detecting both cECG and cough-associated cEMG signals with amplitudes of approximately 100 µV for subject B from the neck via a pillowcase in an environment with a low humidity of 6.10 g/m^3^. Similar results were obtained from seven of eight subjects, and the average R-wave detection accuracy exceeded 85%, as shown in Table 2. In the cEMG signal, an increase in spectral amplitude from 40 to 500 Hz was observed for the recording segment with cough action in all eight subjects. The mean volumetric humidity during the experiment was 6.85 ± 1.85 g/m^3^, and the respective values of relative humidity were lower than 40% for seven of eight subjects. The successful detection of both ECG and EMG signals from the neck is attributable to the following two points: (I) the total number of sensing electrodes was reduced to two by combining the electrode and the FVF for detecting both signals, and (II) the residual space allowed us to arrange the electrodes diagonally along with the approximate vector of the lead II ECG measurement. As the typical electrical axis of the heart for R-waves varies between 30° and 60° with respect to the horizontal axis when frontally measured on the chest [48], the typical axis is a good fit for our diagonal arrangement of posterior neck measurements.

The successful detection of signals with amplitudes of approximately 100 µV via cloth with a thickness of 308 µm is attributed to the following three points: (a) an FVF with ultra-high input impedance was introduced to suppress the voltage loss at the electrode–skin coupling, (b) no ground electrode was used to secure the area for the two sensing electrodes in the limited neck space, and (c) a pillow with an indentation in the neck region was adopted to maintain coupling between the electrodes and the neck. Regarding point (a), the input impedance $Z_{in\_bt}$ of the proposed FVF shown in Figure 2c, is expressed as follows [53]:(5)$$Z_{in\_bt}=R_{1}+R_{bv}+\frac{R_{1}R_{bv}}{R_{3}}.$$

Assuming that the varistor used in this study (TDK, AVR-M1608C080MTAAB) has a resistance $R_{bv}$ of 250 MΩ when the voltage applied to the varistor $v_{vrst}$ is less than the threshold voltage (8 V), $Z_{in\_bt}$ is approximately 2.5 TΩ according to Equation (4) for the resistances $R_{1}$ (99 MΩ) and $R_{3}$ (10 kΩ) shown in Figure 2c. Furthermore, because the input impedance of the OP amplifier used in this study (Texas Instruments, OPA129) is 10 TΩ according to the data sheet, the input impedance $Z_{in\_syn}$ of the analog front-end circuit synthesized in parallel with the bootstrapping component is calculated as approximately 2.0 TΩ. Meanwhile, the coupling capacitance is calculated as 57.5 pF based on C=εS/d with a dielectric constant of $\varepsilon_{r}=\;8.85\times 10^{-12}$ F/m, a coupling area of $S=2.00\times 10^{-3}$ m^2^, and a coupling distance of $d=3.08\times 10^{-4}$ m. Assuming that the dominant frequency band of the ECG R-wave is 10–20 Hz, the capacitive reactance of the coupling $X_{C}=d/\left(2\pi f\varepsilon S\right)$ is 277–138 MΩ. Therefore, a sufficiently large $Z_{in\_syn}$ (>>$X_{C}$) is achieved by the FVF used in this study. Moreover, both points (b) and (c) contributed to the decreased coupling reactance $X_{C}$ associated with an increased coupling area S and decreased coupling distance d. As a result, the voltage loss at the coupling was considered to be suppressed.

The successful detection of both ECG and EMG signals in a low-humidity environment is attributable to the following three points: (i) the adoption of an FVF with a voltage-dependent input impedance, (ii) the adoption of five-layered electrodes with rapid static discharge characteristics [50], and (iii) the adoption of a pillow with an indentation in the neck. Regarding point (i), the varistor has a nonlinear resistance that can be approximated by the following equation:(6)$$R_{bv}\approx \left\{\;\begin{array}{cc} R_{h}, & if\;-V_{thr}<v_{vrst}<+V_{thr}\\ 0, & else \end{array}\right.,$$
where *v_vrst_* is the voltage applied to the varistor and *V_thr_* is the threshold voltage. It has been reported that the bootstrapping component, including the varistor shown in Figure 2c, has a voltage-dependent impedance, as given by Equation (7) [53]:(7)$$Z_{in\_bt}=R_{1}+R_{bv}+\frac{R_{1}R_{bv}}{R_{3}}\approx \left\{\;\begin{array}{cc} \frac{R_{1}}{R_{3}}R_{h\;}, & \;if\;-V_{thr}<v_{vrst}<+V_{thr}\\ R_{1}, & else \end{array}.\right.$$

Because $v_{vrst}=\frac{R_{h}}{R_{h}+R_{3}}v_{in}$, $Z_{in\_bt}$ has a low impedance $R_{1}$ when $v_{in}$ exceeds a threshold $\pm \;V_{thr}\left(1+\frac{R_{3}}{R_{h}}\right)$. As the time constant of the measurement system is the product of the input impedance and coupling capacitance of the measuring site, a smaller decrease in the input impedance of the FVF corresponds to a shorter time constant. Consequently, the discharge duration for static electricity on the body or clothing, which commonly develops under low-humidity conditions, becomes shorter, leading to a stable measurement. As for point (ii), it has been shown that five-layered (i.e., doubly shielded) electrodes have superior discharge characteristics over single- and three-layered electrodes when a 5000-V electrostatic discharge is applied to the electrodes in an electrostatic discharge immunity test (IEC61000-4-2) [50]. Five-layered electrodes are reported to mitigate the motion artifacts caused by an impact hammer [49]. Therefore, it was considered that the electrode configuration mitigated the artifacts induced by cough-associated body movements in the present study, and R-waves were detected after the two-cough cycle (see Figure 5a). Concerning point (iii), the rapid coupling between the electrodes and the neck caused by the indentation may reduce the occurrence of friction between the skin and the pillowcase.

### 4.2. Comparison of Heart Rate Variability between the Proposed and Commercial Systems

Heart rate variability (HRV) analysis of the ECG_ref_ and cECG data was performed on six subjects with a *P_ACC_* of 100% (Table 2). Timing data of the R-waves detected in the ECG analysis of 2.4.1 were analyzed by the “HRV analysis” function in commercially available software (AcqKnowledge4.1, BIOPAC Systems). Spectral power densities (PSDs) of very low frequency (VLF, DC–0.04 Hz), low frequency (LF, 0.04–0.15 Hz), high frequency (HF, 0.15–0.40 Hz), and very high frequency (VHF, 0.40–3.00 Hz) were respectively calculated, and are summarized in Table 3.

As evidenced in Table 3, the maximum PSD error rate between cECG and ECG_ref_ was 0.43%. The mean error rate in each band was less or equal to 0.2%. Therefore, the HRV values were measured accurately in all six subjects during the resting period. The next step is validation of the system in long-duration measurements.

### 4.3. Challenges to Be Tackled

As this study was pursued as a proof-of-concept, the method has limited applicability to real scenarios. We note that the accuracy of R-wave detection in subject E was below 15% (see Table 2), and the cECG recording in Figure 6 revealed a more distorted waveform, with the R-wave having a smaller amplitude than the T-wave. Because the frequency components of R-waves are generally larger than those of T-waves, the observed distortion (i.e., large attenuation at higher frequencies) is attributable to stray capacitance between the lead wire of each sensing electrode and the circuit ground, as reported in [43]. Nakamura et al. recently reported on a novel analog front-end circuit that exhibited characteristics of stray capacitance reduction. Therefore, the substitution of our FVF for this analog front-end circuit would mitigate the distorted attenuation observed in the cECG waveform. The recording in Figure 6 also implies that the small amplitude of the original R-wave in subject E caused the low detection ratio and that noise reduction can improve the R-wave detection accuracy. So-called driven shield ground (DSG) is known to suppress common-mode noise such as power line noise [55] in capacitive biopotential measurements. Thus, the introduction of a DSG may enhance the detection accuracy for subjects with small R-wave amplitudes. A DSG also suppresses common-mode components in motion artifacts [55]. Therefore, introducing a DSG might further improve the detection of R-waves during or after the cough-associated body movements.

In addition to the above described hardware approaches, the application of a more sophisticated algorithm (e.g., publicly available toolboxes [56]) instead of the current threshold approach may enhance the detectability of R-waves.

For patients with a persistent cough, the total number of coughs at night can exceed 100 [28]. To apply the proposed system to these patients, an algorithm is needed to count the coughs in a recorded signal. However, the proposed system detected cEMG signals associated not only with coughs but also with inhalation and/or swallowing in the experiment, as shown in Figure 9. Therefore, an algorithm for distinguishing among coughs, inhalation, and swallowing would be important for clinical applications. Aside from electromyographic interference, mechanical vibrations from speech or sound did not affect the outputs from the electrodes, as evidenced by the time series waveforms and their DFT spectra (data not shown).

The configuration and dimensions of the current electrode narrow the applicable subjects and situations. The subjects include infants and bedridden elderly with thin hair who do not roll over. To widen the acceptance of the device, the measurements of cECG and cEMG in lateral postures through thick hair should be attained in a wide age range of the subjects.

## 5. Conclusions

In this study, the authors developed a proof-of-concept non-contact system for performing cECG and cough-associated cEMG measurements using in-pillow electrodes. The sensing electrodes were arranged under a pillow cover so that the vector between the electrodes was similar to that of lead II ECG. The same sensing electrodes were used for the cEMG measurement. To detect biopotentials with amplitudes of approximately 100 µV, a signature analog front-end circuit was introduced in the measurement system. A bootstrapping technique and a nonlinear electrical component (varistor) were combined to attain high input impedance and rapid static discharge characteristics. cECG and cEMG signals were measured with a 308-µm-thick pillow cover on the fabricated electrodes for seven adult males and one female (average age: 22.5 ± 1.3 yr) during resting or coughing. The accuracy of R-wave detection for a 2-min resting period was 100% in six subjects, with an average of 87.5% ± 30.0% for all subjects. Clearly visible cEMGs were measured for all 14 cough motions in each subject, corresponding to the reference EMGs obtained from submental muscle. The power spectra of non-contact cEMGs during coughing showed an increase in 40–500 Hz compared with that obtained during resting. Although its practical applicability could be improved, the proposed method is a potentially useful method for routine measurement for ECG and cough-associated EMG through cervical pillows.

Future work includes the introduction of a negative capacitance circuit and a bandpass amplifier at the analog front-end to improve the accuracy of R-wave detection for low signal-to-noise ratios, such as that observed in subject E. In addition, a more sophisticated algorithm is to be introduced for the improvement of the accuracy of R-wave detection.

## Figures and Tables

**Figure 1 sensors-21-00812-f001:**
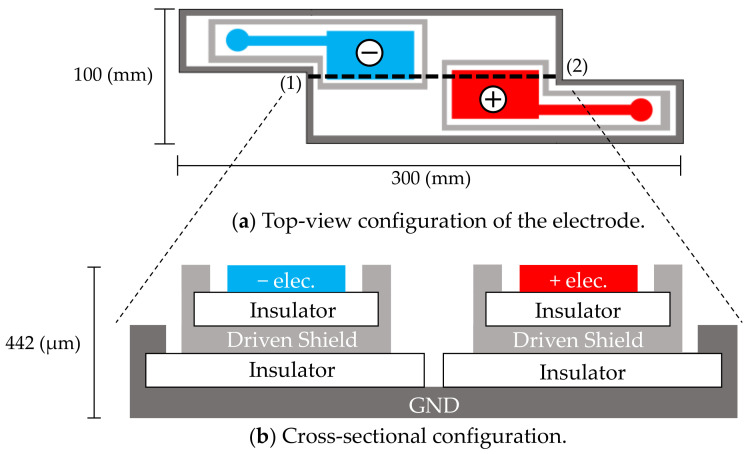
In-pillow capacitive cloth electrodes. (**a**) Top-view configuration of the electrodes. (**b**) Cross-sectional configuration of (**a**) along the dotted line (1)–(2). The electrodes have a five-layered structure. The top layers consist of a negative electrode (−elec.) and a positive electrode (+elec.).

**Figure 2 sensors-21-00812-f002:**
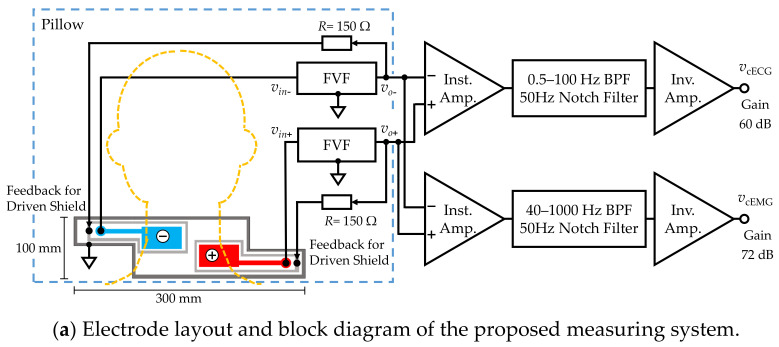
Proposed measuring system. (**a**) Electrode layout and block diagram. (**b**) Function of front-end voltage follower (FVF) in capacitive biopotential measurements (**c**) Circuit diagram of the FVF employed in the measuring system. A bootstrapping technique and varistor element were adopted to increase the voltage-dependent input impedance.

**Figure 3 sensors-21-00812-f003:**
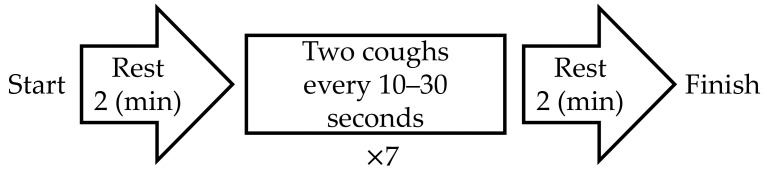
Protocol of experiment in this study.

**Figure 4 sensors-21-00812-f004:**
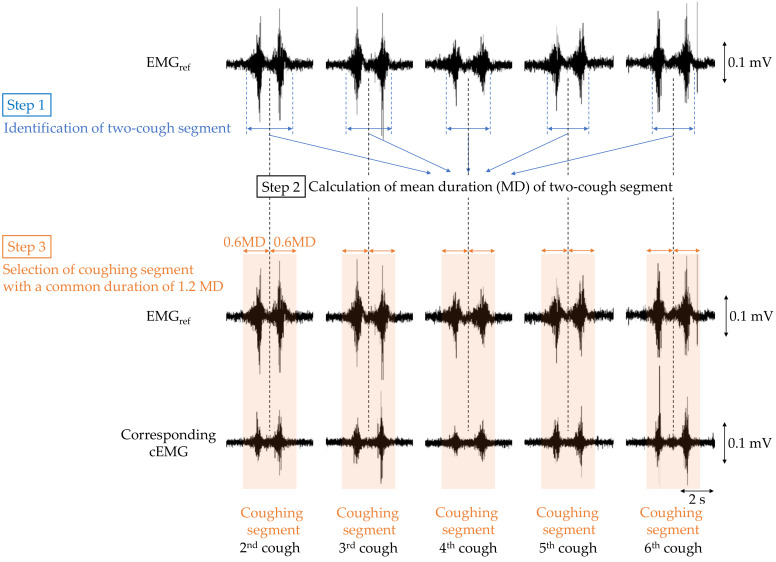
Flowchart of selecting the coughing segment in the EMG_ref_ and cEMG data for the following discrete-time Fourier transform (DFT) analysis.

**Figure 5 sensors-21-00812-f005:**
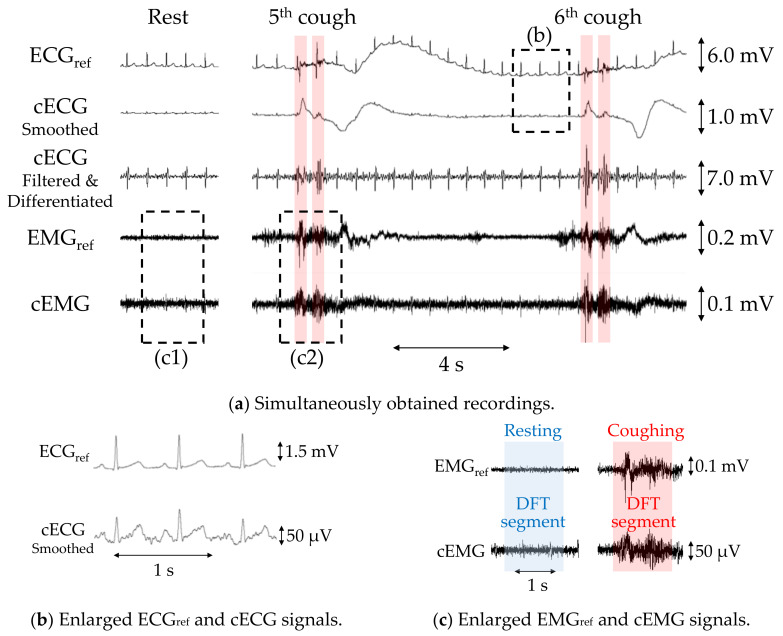
Simultaneous measurements of subject B with the proposed or commercial system obtained during a two-cough experiment. (**a**) Recordings of ECG_ref_, cECG, differentiated cECG, EMG_ref_, and cEMG. (**b**) Enlarged ECG_ref_ and cECG waveforms and (**c**) enlarged EMG_ref_ and cEMG signals corresponding to the dotted rectangular sections in (**a**).

**Figure 6 sensors-21-00812-f006:**
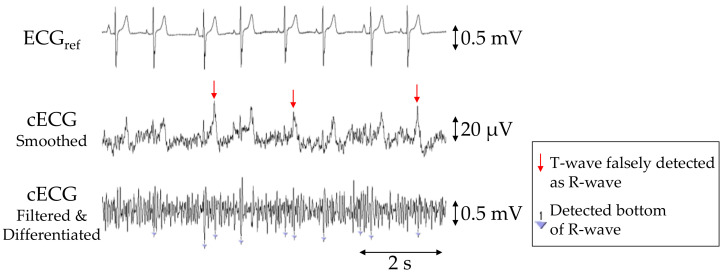
Simultaneous recordings of ECG_ref_, cECG and filtered and differentiated cECG of subject E obtained with the commercial system (first stage) or the proposed system (second and third stages) during the 2-min segment before the first two-cough experiment. Inverted triangles marks in the third stage show the detected bottoms of the R-waves.

**Figure 7 sensors-21-00812-f007:**
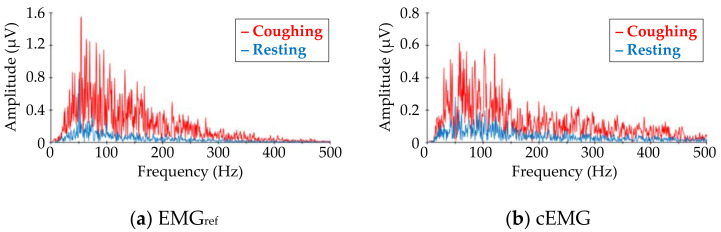
Comparison of frequency–amplitude spectra between coughing and resting segments for (**a**) EMG_ref_ and (**b**) cEMG data shown in Figure 5c. The red line corresponds to the spectra for coughing segments, while the blue line corresponds to that for resting segments. (Subject #B).

**Figure 8 sensors-21-00812-f008:**
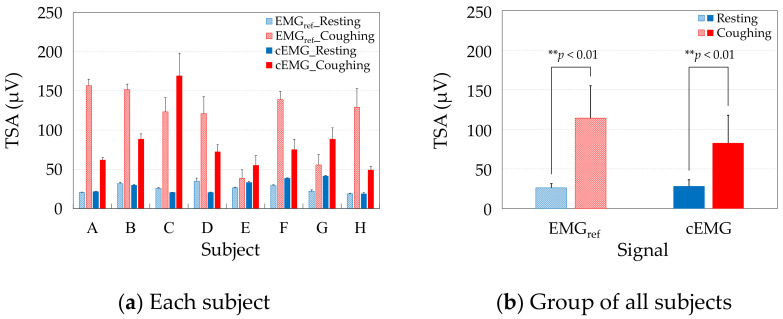
Comparison of mean total spectral amplitudes (TSAs) between resting and coughing in the EMG_ref_ and cEMG signals of (**a**) each subject, and (**b**) the group of all subjects.

**Figure 9 sensors-21-00812-f009:**
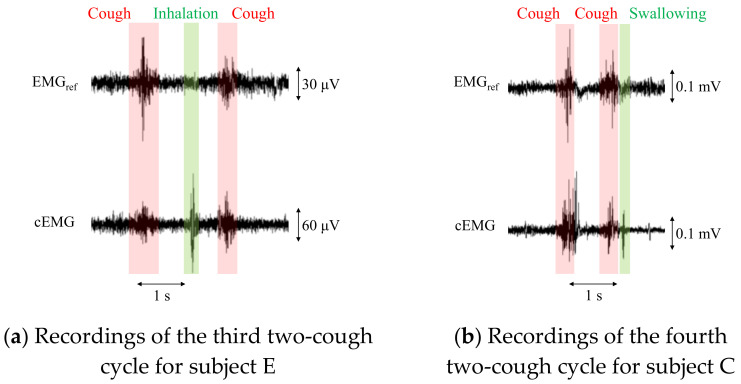
Recordings of EMG_ref_ and cEMG including a two-cough cycle for subjects E and C. In the cEMG signals, firings considered to be associated with inhalation (**a**) and swallowing (**b**) were also confirmed by the two-cough procedure.

**Table 1 sensors-21-00812-t001:** Subject information and experimental conditions.

Subject	Weight (kg)	Height (m)	BMI (kg/m^2^)	Age (yr)	Temp (°C)	RH (%)	VH (g/m^3^)
A	74	1.61	28.5	23	23.5	36.0	7.62
B	60	1.75	19.6	23	24.0	28.0	6.10
C	70	1.63	26.3	23	23.8	33.0	7.11
D	48	1.53	20.5	20	23.8	43.8	9.44
E	57	1.66	20.7	24	21.8	28.0	5.38
F	90	1.65	33.1	23	23.6	35.0	9.34
G	64	1.61	24.7	23	25.2	22.0	5.13
H	77	1.80	23.8	21	23.5	22.0	4.66
Mean ± SD	67.5 ± 13.1	1.7 ± 0.1	24.6 ± 4.6	22.5 ± 1.3	23.7 ± 0.9	31.0 ± 7.5	6.85 ± 1.85

BMI: body mass index, Temp: temperature, RH: relative humidity, VH: volumetric humidity.

**Table 2 sensors-21-00812-t002:** Sensitivity (*P_SNS_*) (%), accuracy (*P_ACC_*) (%), and positive predictive value (*P_PPV_*) (%) of R waves.

Subject	*P_SNS_* (%)	*P_ACC_* (%)	*P_PPV_* (%)	*N_TP_ + N_FN_*
A	100	100	100	163
B	100	100	100	174
C	100	100	100	172
D	92.5	85.4	91.8	133
E	33.1	14.3	20.2	118
F	100	100	100	141
G	100	100	100	122
H	100	100	100	135
Mean ± SD	90.7 ± 23.4	87.5 ± 30.0	89.0 ± 27.9	145 ± 22

**Table 3 sensors-21-00812-t003:** Comparison of spectra of spectral power density (PSD) of heart rate variability (HRV) between ECG_ref_ and cECG.

Subject	VLF ×10^−4^ (s^2^/Hz)		Error Rate (%)	LF ×10^−4^ (s^2^/Hz)		Error Rate (%)	HF ×10^−4^ (s^2^/Hz)		Error Rate (%)	VHF ×10^−4^ (s^2^/Hz)		Error Rate (%)
	ECG_ref_	cECG		ECG_ref_	cECG		ECG_ref_	cECG		ECG_ref_	cECG	
A	8.9283	8.9286	0.004	62.607	62.615	0.014	16.102	16.098	0.02	8.539	8.531	0.09
B	8.0537	8.0543	0.008	54.646	54.645	0.002	14.659	14.657	0.02	7.804	7.811	0.10
C	8.1603	8.1624	0.026	56.258	56.273	0.027	15.446	15.435	0.08	7.714	7.747	0.43
F	11.9863	11.986	0.002	82.874	82.879	0.006	23.857	23.815	0.17	11.395	11.415	0.17
G	15.3784	15.378	0.000	105.311	105.247	0.061	29.195	29.141	0.19	15.340	15.299	0.27
H	13.2398	13.244	0.028	88.198	88.259	0.070	23.870	23.837	0.14	12.350	12.333	0.14
Mean	10.9578	10.9589	0.011	74.982	74.986	0.030	20.521	20.497	0.10	10.524	10.523	0.20

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
