# Peer review of "Non-Contact Measurements of Electrocardiogram and Cough-Associated Electromyogram from the Neck Using In-Pillow Common Cloth Electrodes: A Proof-of-Concept Study"

_sensors, 2021, doi:10.3390/s21030812_

Round 1

Reviewer 1 Report

The paper presents a system based on capacitively-coupled conductive-tissue electrodes for recording EMG and ECG during a patient sleep.

The proposal is interesting, but I see a major shortcoming for practical applicability of the system.
It seems that the total electrode dimension is fairly small (10 cm) and therefore the patient head should be placed in a very specific and rather restrained posture with reference to the pillow. I think this is impractical, since everybody will change posture several times during sleep. How do the authors imagine to tackle this problem?

Moreover, the proposed design contains several solutions which are either poorly described, or whose effect on improving the performance of the system remain undemonstrated.
Just a couple of examples follow.

  1. The authors greatly emphasize that the electrodes have been arranged along the lead II ECG vector. I don't understand why this specific orientation should provide any advantage, considering that the two electrodes are placed both on the patient neck (which is not the correct location for lead II ECG) and, as highlighted before, the posture of the head relative to the pillow might be largely random in a realistic scenario. I would have liked to see a comparison with measurements performed with the two electrodes in a different orientation.
  2. The designed electrodes exhibit a "driven shield". The exact role of this shield is not fully clear and, above all, the driving circuit is described nowhere. Once again: what would be the result of removing (or simply leaving "undriven") this shield?

Finally, I also have some doubts about the suitability of the adopted metrics to assess the actual performance of the system as refers to ECG acquisition. All the proposed parameters are aimed at assessing the capability of correctly detecting R waves. This is already a good indicator, but the authors stress that the final aim is to extrapolate HR variability. I would have also liked to see some comparison between HR variability-linked parameters detected with the reference and the capacitive systems.

Reviewer 2 Report

The authors propose a non-contact system for performing capacitive ECG, for the obtention of HR and HRV, and cough-associated capacitive EMG measurements using cloth electrodes under a pillowcase. Results show a good performance of the proposed sensors in comparison with reference devices. In the overall, the topic is of scientific interest and the manuscript is well written. To improve the quality of the manuscript, some clarifications on the methods for evaluation are required as well as some more robust techniques are suggested.

Comments:

  • “For R-wave detection, 80% of each mean amplitude was used as the threshold through the segment.” Does that mean you used a threshold approach to obtain R peaks? Why did not you use a more robust automatic algorithm like the ones publicly available (e.g. http://biosig.sourceforge.net/)?  This could help a fairer evaluation of the cECG signal quality. In fact, it could enhance the detection of R waves in subject E.
  • “For each two-cough cycle in the EMGref signal, the segment… and its period were identified”. What do you mean with “its period”? This processing approach would benefit from a figure to exemplify it.
  • In general, the method utilized for the validation of EMG segments is unclear. One would think that running a on-off EMG detector for both the cEMG and the EMGref could help identify if the cEMG signals are successful in detecting cough events, and how many false positives or negatives are detected. Spectral features off the EMG segments could also be computed, including SMR, SNR, etc. (for example, see DOI: 10.1109/JTEHM.2016.2567420). Spectral correlation could also be computed. Obtaining only segments of cEMG correspondent to events detected in the EMGref (if this is what is being done), is inappropriate. Please explain better and argue about your approach.
  • As they are capacitive, are those electrodes able to sense sound? If so, this could be analogous to using a microphone and the privacy issue remains.
  • If the ECG signals are only meant for R peak detection (HR and HRV) and none of the other waves are relevant, a lower cut-off frequency could be considered (0.5 – 50 Hz, for instance).
  • Any comments on how are those electrodes performing in presence of motion artifacts?
  • A brief explanation of the functioning of the circuit without the bootstrapped varistor should be added, to better explain why it was incorporated. I appreciate some details are provided in the discussion section.
  • I suggest adding a paragraph with the limitations of the study and the contactless system itself. For example, as you have said you controlled the position of the hair which is unrealistic in a real scenario.

Round 2

Reviewer 1 Report

Thank you for having considered all the comments and modified the manuscript accordingly.

I have no further suggestions.

Reviewer 2 Report

The authors have addressed my comments and suggestions. The new manuscript is a better product.